# Generalized Parallel Scaling with Interdependent Generations

**Harry Dong**[1,2*] **David Brandfonbrener**[1] **Eryk Helenowski**[1] **Yun He**[1] **Mrinal Kumar**[1]
**Han Fang**[1] **Yuejie Chi**[1,3] **Karthik Abinav Sankararaman**[1]
[1]Meta
[2]Carnegie Mellon University
[3]Yale University
harryd@andrew.cmu.edu

## ABSTRACT

Parallel LLM inference scaling involves sampling a set of $N > 1$ responses for a single input prompt. However, these $N$ parallel responses tend to be generated independently from each other, partitioning compute resources and leaving potentially useful information in one generation untapped by others. This is in contrast to response length scaling where past computation is used in all future steps. For higher quality responses and response sets, we propose `Bridge` to generate *interdependent responses in parallel* by rethinking batched LLM hidden states as holistic tensors rather than independent slices. With only a small amount (2.8%-5.1%) of new parameters, `Bridge` improves the relative mean accuracy gains from reinforcement learning with verifiable rewards by up to 39% and boosts consistency of correct responses. Trained once, `Bridge` scales to any generation width, all with greater performance than independent generations, unlocking a more general mode of parallel scaling that effectively leverages information between sequences, compatible with any post-generation aggregation technique.

## 1 INTRODUCTION

Scaling inference-time compute has given large language models (LLMs) substantial leaps in performance on difficult tasks. Many scaling methods concentrate resources to generate a *single* high-quality response such as with chains-of-thought (CoTs) (Wei et al., 2022) and decompositions of a problem into parallel substeps (Rodionov et al., 2025; Yang et al., 2025b). However, there are also instances where a high-quality *set* of responses for each input is needed, such as in the case of output synthesis, best-of-$N$ selection, and synthetic data generation. Scaling this in a parallel manner is traditionally done by Monte Carlo sampling independent generations. Consequently, each generation is ignorant of the other rollouts, despite answering the same prompt. Independent generations for the same prompt leave potentially useful information derived from other responses unutilized, limiting the performance ceiling. In contrast, sequentially scaling CoTs ensures each sampled token can play a role in the final output. Motivated by the potential of shared information across parallel generations stemming from the same prompt, we aim to leverage these interactions to enhance and generalize parallel inference scaling.

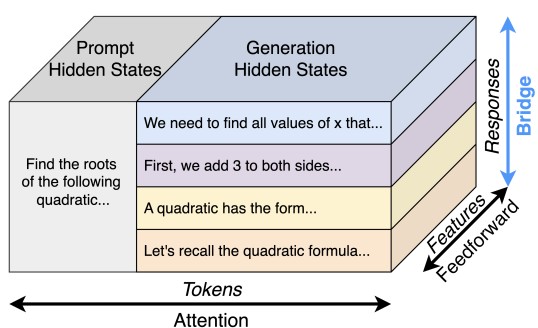

Figure 1: LLM hidden states are 3-D tensors, where attention and feedforward blocks explicitly transfer information between tokens and features, respectively. By instead treating parallel scaling generations as a single tensor rather than independent slices, our method, `Bridge`, operates along the batch axis, so that tokens from all sequences that share the same prompt can share information throughout generation.

---

*Work done during an internship at Meta

There has been progress in integrating parallel dependence for inference, such as breaking down reasoning steps into parallel paths (Rodionov et al., 2025; Pan et al., 2025a; Jin et al., 2025; Yang et al., 2025b). There, parallel computation is funneled into a single output, useful for generating one response but not a *set* of responses. Even so, they highlight the potential of mid-generation interactions between sequences. We seek to extend parallel scaling with *interdependence*, which allows all $N$ output sequences for one prompt to use all the compute and information available, not just a single isolated partition. *Thus, the challenge is finding a totally parallel method that uses $N$ simultaneous threads to generate $N$ responses with interdependence without extensive post-training.*

Looking at LLM hidden states operations reveals a clue to overcome these challenges (Figure 1). For batch size $B$, sequence length $S$, and hidden dimension $D$, the hidden states per forward pass has 3-D shape $B \times S \times D$. Attention and feedforward blocks blend information throughout each $S \times D$ slice with the batch axis kept independent. Even minor inter-sample interactions like Batch Normalizations (Ioffe & Szegedy, 2015) were substituted with Layer Normalizations (Ba et al., 2016). While this is natural for heterogeneous batches where samples with wildly different inputs can be fed together without interference, *parallel scaling, which draws many responses from a single input, exhibits uniquely homogeneous structure as each output stems from the same input.* Hence, there is the potential for useful information transfer during the generation process which we exploit.

We introduce `Bridge` (**B**atch **r**easoning with **i**nter**d**ependent **ge**nerations), a method that shares information across tokens that stem from the same prompt in a batch for parallel scaling with interdependent generations. With a minor architectural change to LLMs, each token generated in a batch can depend on tokens in other generation threads with the same prompt. In turn, our method improves reasoning performance evaluated both at the individual response level (accuracy) and response set level (coverage and G-Pass@$k_\tau$ (Liu et al., 2025)). Furthermore, our method focuses on generation, so any post-generation aggregation technique can be used. We push the following advancements for parallel scaling:

1. **Parallelism with Dependence:** Instead of generating in isolated silos, `Bridge` allows information to flow between sequences while maintaining complete generation parallelism. Thus, inference compute is pooled together for all tokens, rather than being partitioned. `Bridge` significantly increases the final performance after reinforcement learning with verifiable rewards (RLVR) on 12 benchmarks using multiple reasoning models.

2. **Low Cost:** By adding only 2.8% to 5.1% additional parameters, and warming up on a small supervised fine tuning (SFT) dataset (e.g. GSM8K (Cobbe et al., 2021)), `Bridge` already significantly improves the effectiveness of RLVR.

3. **Versatility:** `Bridge` has no restriction on the width of parallelism and is robust to train-time and test-time width discrepancies. Trained once, all tested widths outperform independent generations in terms of accuracy, coverage, and consistency. Furthermore, `Bridge` does not rely on any heuristics or interventions at any point in the generation process.

Our extensive experiments on multiple models and tasks show that `Bridge` effectively shares information across multiple generations for the same input. For example, our method improves the relative benefit of RLVR on DeepSeek-R1-Distill-Qwen-7B by 39% averaged over 7 math tasks, compared to the next best method. With the same model, `Bridge` also increases the rate at which all responses to a single competition math problem are correct from 15.0% to 17.8%.

**Paper Organization.** In Section 2, we cover relevant background on test-time scaling with an emphasis on parallel scaling. Then, we introduce `Bridge` in Section 3, detailing the algorithm, the training pipeline, and its implications. We demonstrate `Bridge`'s efficacy on a variety of reasoning datasets, evaluated both on sample-wise accuracy and on global response set quality in Section 4. We go further and provide a thorough investigation of our method including varying the generation width, sequence length extrapolation, learned features, and output analysis in Section 4.3.

## 2 BACKGROUND & RELATED WORKS

We start with an overview of test-time scaling methods for LLMs, emphasizing parallel methods. Although some methods combine parallel generations into one response, the problem of generating a high-quality set of interdependent responses, which we aim to tackle, remains understudied.

**Test-time Scaling.**    In part due to the success of scaling LLM training (Kaplan et al., 2020; Hoffmann et al., 2022), there has been a growing interest in quantifying how far scaling LLM inference-time compute can push performance, especially on difficult reasoning tasks. The main axes of inference scaling are generation length and the number of generations. To scale generation length, LLMs are encouraged to produce long CoTs before arriving at a final answer (Guo et al., 2025; Yang et al., 2025a; Muennighoff et al., 2025) which is usually of higher quality than shorter CoTs. In the case of extreme generation lengths, there appears to be diminishing or even negative returns, suggesting a limitation of current models to scale indefinitely along this axis (Gema et al., 2025). To scale along the number of generations axis, LLMs can output multiple responses for a single query, increasing the probability of a high quality response being generated (Brown et al., 2024; Snell et al., 2024; Wu et al., 2024; Manvi et al., 2024; Sun et al., 2024; Dong et al., 2025). However, independent generations divide computational resources among themselves, oblivious to each other's progress, which leads to less significant performance gain with additional compute than length scaling (Mirtaheri et al., 2025). To promote parallel exploration during training, training with a Pass@$k$ objective has shown promise which could be an interesting extension to our method (Chen et al., 2025b).

**Post-generation Synthesis.**    Instead of selecting one response from a pool of candidate responses, some works investigate ways to synthesize multiple responses together. One way is to take an unweighted or weighted majority vote across responses (Wang et al., 2022; Uesato et al., 2022; Lightman et al., 2023; Li et al., 2023), but this is geared mainly for discrete answers, and an effective synthesis of reasoning traces remains unclear. There are also approaches where multiple responses are concatenated and fed into an LLM to extract or combine information (Chen et al., 2023; Qi et al., 2025; Zhao et al., 2025). *Our work's focus is on the generation phase, so many of these post-generation aggregation techniques can be seamlessly integrated.*

**Mid-generation Synthesis/Pruning.**    There has also been some work in developing techniques to share information across outputs mid-generation. For instance, Hogwild! Inference (Rodionov et al., 2025) and Group Think (Hsu et al., 2025) share key-value caches across generation runs to collaborate and decompose tasks into subtasks. Similarly, some methods concatenate outputs of parallel processes to aid in the main decoding thread (Pan et al., 2025a; Jin et al., 2025; Yang et al., 2025b; Macfarlane et al., 2025). Training from scratch, ParScale (Chen et al., 2025a) fans outs an input into multiple paths within the model architecture then aggregates them to predict the next token. Whereas these previous methods funnel resources of $N$ parallel processes to produce one output, our method uses $N$ parallel processes to simultaneously generate $N$ high quality outputs. This way, our design integrates inter-output dependency mid-generation while producing different responses simultaneously, flexibly suitable for post-generation synthesis, RLVR training, and synthetic data generation. Another line of work involves stopping unpromising outputs mid-generation to devote more resources to other parallel generations (Fu et al., 2025; Sun et al., 2024). These works show excellent reductions in compute, and composing them with our method is of interest for future work.

**High Order Tensors.**    Multiple works that have observed richer representations in tensors than their flattened matrix counterparts (Kolda & Bader, 2009; Papalexakis et al., 2016). Converting this observation into tractable algorithms remains a major area of research in theory (Tong et al., 2022; Dong et al., 2023c; Luo & Zhang, 2023; Qin et al., 2025) and practice (Ho et al., 2019; Dong et al., 2023b). We take inspiration from these works to diffuse information throughout the entire LLM hidden states tensor instead of being constrained to matrix slices.

## 3    BRIDGE: CONNECTING GENERATION PATHS

Sharing information between samples mid-generation in the latent space gives rise to a couple technical challenges. One is finding an effective and efficient way to achieve this. Attention and feedforward blocks already pose serious static and dynamic memory bottlenecks, which we want to avoid accentuating while still improving accuracy. Second, we also need versatility to allow for any number of parallel generations at test-time. `Bridge` overcomes these challenges with small attention-like blocks that fit into any LLM. We begin with a description of `Bridge`, its connections, and its implications in Section 3.1, followed by SFT warm up (Section 3.2) and RLVR (Section 3.3) details.

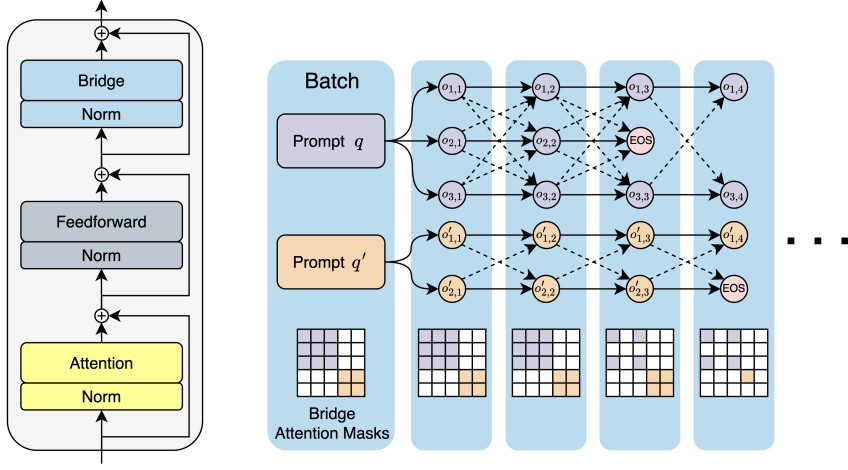

Figure 2: Our method design. **(Left)** A `Bridge` block and input normalization layer are added after each feedforward block. **(Right)** A timestep's tokens stemming from the same input prompt attend to each other in `Bridge` blocks, denoted by the arrows. Dotted arrows illustrate all the locations of information transfer to different sequences in a Markovian fashion (token features only at the current timestep are shared to predict the next timestep's tokens). Attention is masked for tokens from different prompts and from completed generations. White squares are masked cells.

## 3.1 BRIDGE ARCHITECTURE

We introduce `Bridge`, a new transformer (Vaswani et al., 2017) block that introduces dependence between samples in a batch. At a high level, `Bridge` performs attention between tokens, which share the same prompt and do not come from completed generations, in a batch at each timestep. We summarize our method in Figure 2 and Algorithm 1.

To start, we first describe self-attention layers. Define hidden states $\boldsymbol{\mathcal{X}} \in \mathbb{R}^{B \times S \times D}$ for batch size $B$, sequence length $S$, and hidden dimension $D$. Let $[\boldsymbol{\mathcal{X}}]_{b,\cdot,\cdot}$ and $[\boldsymbol{\mathcal{X}}]_{\cdot,s,\cdot}$ be the $b$-th and $s$-th 2-D slices along the batch and sequence axes, respectively. Self-attention, parameterized by $\boldsymbol{W}_{\mathrm{A,Q}}, \boldsymbol{W}_{\mathrm{A,K}} \in \mathbb{R}^{D \times D_{\mathrm{QK}}}$ and $\boldsymbol{W}_{\mathrm{A,V}}, \boldsymbol{W}_{\mathrm{A,O}}^{\top} \in \mathbb{R}^{D \times D_{\mathrm{VO}}}$, is calculated independently for each sample $b$:

$$\boldsymbol{Q}_{\mathrm{A},b} = [\boldsymbol{\mathcal{X}}]_{b,\cdot,\cdot}\boldsymbol{W}_{\mathrm{A,Q}}, \qquad \boldsymbol{K}_{\mathrm{A},b} = [\boldsymbol{\mathcal{X}}]_{b,\cdot,\cdot}\boldsymbol{W}_{\mathrm{A,K}}, \qquad \boldsymbol{V}_{\mathrm{A},b} = [\boldsymbol{\mathcal{X}}]_{b,\cdot,\cdot}\boldsymbol{W}_{\mathrm{A,V}},$$

$$[\mathrm{Attn}(\boldsymbol{\mathcal{X}})]_{b,\cdot,\cdot} = \underbrace{\mathrm{Softmax}(\mathrm{Mask_A}(\boldsymbol{Q}_{\mathrm{A},b}\boldsymbol{K}_{\mathrm{A},b}^{\top}))}_{\in \mathbb{R}^{S \times S}} \boldsymbol{V}_{\mathrm{A},b}\boldsymbol{W}_{\mathrm{A,O}}. \tag{1}$$

`Bridge` blocks are similar, but attention between samples is calculated independently for each token index $s$. Letting $\boldsymbol{W}_{\mathrm{B,Q}}, \boldsymbol{W}_{\mathrm{B,K}} \in \mathbb{R}^{D \times D_{\mathrm{QK}}}$ and $\boldsymbol{W}_{\mathrm{B,V}}, \boldsymbol{W}_{\mathrm{B,O}}^{\top} \in \mathbb{R}^{D \times D_{\mathrm{VO}}}$,

$$\boldsymbol{Q}_{\mathrm{B},s} = [\boldsymbol{\mathcal{X}}]_{\cdot,s,\cdot}\boldsymbol{W}_{\mathrm{B,Q}}, \qquad \boldsymbol{K}_{\mathrm{B},s} = [\boldsymbol{\mathcal{X}}]_{\cdot,s,\cdot}\boldsymbol{W}_{\mathrm{B,K}}, \qquad \boldsymbol{V}_{\mathrm{B},s} = [\boldsymbol{\mathcal{X}}]_{\cdot,s,\cdot}\boldsymbol{W}_{\mathrm{B,V}},$$

$$[\mathtt{Bridge}(\boldsymbol{\mathcal{X}})]_{\cdot,s,\cdot} = \underbrace{\mathrm{Softmax}(\mathrm{Mask_B}(\boldsymbol{Q}_{\mathrm{B},s}\boldsymbol{K}_{\mathrm{B},s}^{\top}))}_{\in \mathbb{R}^{B \times B}} \boldsymbol{V}_{\mathrm{B},s}\boldsymbol{W}_{\mathrm{B,O}}. \tag{2}$$

There are 3 key differences between usual self-attention and `Bridge` beyond a transposition of $\boldsymbol{\mathcal{X}}$:

- Instead of a decoder mask, `Bridge` applies an attention mask that omits attention to tokens from sequences stemming from different prompts and sequences that have completed generation. See Figure 2 for an example.
- No positional encoding is used to preserve sample position invariance.
- Without attention to previous tokens, `Bridge`'s Markovian design *does not maintain a key-value cache*.

We place a `Bridge` block after each feedforward block with a residual stream and input normalization layer that mimics existing blocks, shown in Figure 2. `Bridge` is active during the prefill stage too, but since all hidden states for the same input are identical, `Bridge` blocks act as linear layers.

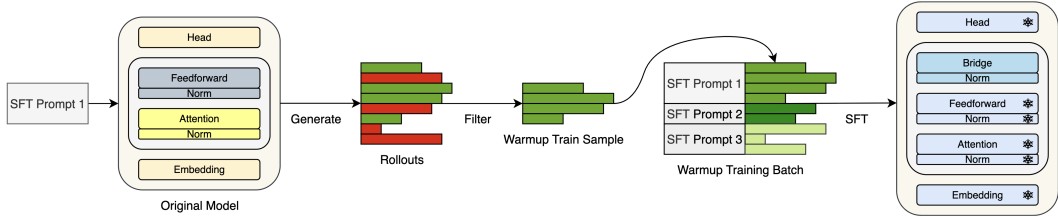

Figure 3: Warm up procedure. The original LLM generates candidate traces which are filtered by correctness and compiled into a dataset. SFT on this generated dataset only updates new parameters. The P-Match baseline substitutes `Bridge` blocks with MLPs matched in parameter count.

**Connection to Efficient Attention for Tensors.** `Bridge` unlocks the ability for an LLM to treat a batch of LLM hidden states as a 3-D ($B \times S \times D$) structure rather than a stack of independent 2-D slices. In this way, the inputs are analogous to images, and the decoding process is like autoregressively generating additional columns. With this interpretation, `Bridge` applying attention operations on different axes of an input is similar to axial attention (Ho et al., 2019) which was introduced first in computer vision to accelerate encoder attention but has since seen wide success in various applications such as in medicine (Azad et al., 2024), materials science (Dong et al., 2023a), and algorithm discovery (Fawzi et al., 2022).

**Generation Interdependence.** For $B$ independent rollouts we sample the next token $o_{b,s+1}$ from

$$p(o_{b,s+1}|q, o_{b,1:s})$$

for sample $b$, timestep $s$, input prompt $q$, and previously generated tokens $o_{b,1:s}$. With `Bridge`, the next token distribution becomes

$$p(o_{b,s+1}|q, \{o_{b',1:s}\}_{b'=1}^{B})$$

for each sample $b$. Conditioned on past tokens, `Bridge` preserves independence between tokens at the same timestep, which allows next token sampling to still be performed in parallel:

$$(o_{b_1,s+1} \perp\!\!\!\perp o_{b_2,s+1})|\{o_{b',1:s}\}_{b'=1}^{B} \text{ for } b_1 \neq b_2.$$

### 3.2 SFT WARM UP

While RLVR can be immediately applied with `Bridge` since these new blocks are initialized to have no contribution, we can also optionally warm them up with SFT for more sufficient training and better downstream performance. A desirable SFT dataset would include many reasoning traces to one prompt. To stay close to the original LLM's generation distribution, we create SFT datasets by first responding to prompts from an existing math dataset. Then, traces are filtered for correctness. During training, these correct traces are fed together in the same batch to warm up `Bridge` blocks with SFT. All other parameters are frozen. Figure 3 illustrates the warm up procedure, and Table 4 explores more in-depth on the benefits of warm up.

### 3.3 RLVR OBJECTIVE

We train LLMs with `Bridge` using GRPO (Shao et al., 2024). We use a variant specified by Yu et al. (2025) which performs token-level normalization to reduce length bias. Letting the group size be $G$, the advantage of the $i$-th output $o_i$ to input $q$ with reward $r_i$ is $\hat{A}_i = \frac{r_i - \text{mean}(r_1,...,r_G)}{\text{std}(r_1,...,r_G)}$. Then, for clipping threshold $\epsilon$, hyperparameter $\beta$, and policy $\pi_\theta$ parameterized by $\theta$, the objective is

$$\mathcal{J}(\theta) = \frac{1}{\sum_{i=1}^{G} |o_i|} \sum_{i=1}^{G} \sum_{s=1}^{|o_i|} \left\{ \min\left[ R_{i,s}(\theta)\hat{A}_i, \text{clip}(R_{i,s}(\theta), 1-\epsilon, 1+\epsilon)\hat{A}_i \right] - \beta D_{\text{KL}}(\pi_\theta || \pi_{\theta_{\text{ref}}}) \right\},$$

$$(3)$$

where

$$R_{i,s}(\theta) = \frac{\pi_\theta(o_{i,s}|q, \{o_{j,1:s-1}\}_{j=1}^{G})}{\pi_{\theta_{\text{old}}}(o_{i,s}|q, \{o_{j,1:s-1}\}_{j=1}^{G})},$$

$$D_{\mathrm{KL}}(\pi_\theta || \pi_{\theta_{\mathrm{ref}}}) = \frac{\pi_{\theta_{\mathrm{ref}}}(o_{i,s}|q, \{o_{j,1:s-1}\}_{j=1}^G)}{\pi_\theta(o_{i,s}|q, \{o_{j,1:s-1}\}_{j=1}^G)} - \log \frac{\pi_{\theta_{\mathrm{ref}}}(o_{i,s}|q, \{o_{j,1:s-1}\}_{j=1}^G)}{\pi_\theta(o_{i,s}|q, \{o_{j,1:s-1}\}_{j=1}^G)} - 1.$$

The key differences from the GRPO objective (and its variants) and our objective are not formulaic but rather inherently induced from the architecture of `Bridge`. Namely, the ratio and KL divergence terms now contain inter-sample dependence between relevant samples, breaking the original assumption of independent trajectories. By linking the advantages and logits in a group, the loss and gradients per output are intertwined with other outputs' that share the same prompt. In other words, gradients from all sequences, containing both positive and negative advantages, are backpropagated through each sequence because of `Bridge` blocks. Further considerations with this setup are discussed in Appendix C. Since `Bridge` is just an architectural change, training is not just limited to SFT and RLVR for reasoning problems. For instance, `Bridge` may also be applied for reinforcement learning from human feedback (RLHF) which is an interesting future direction.

## 4 EXPERIMENTS

We now showcase the benefit of `Bridge` across multiple models and math reasoning benchmarks. After describing our setup in Section 4.1, we first show that applying RLVR with `Bridge` blocks improves accuracy more than other methods. For instance, DeepSeek-R1-Distill-Qwen-7B with `Bridge` blocks observes a relative 39% further improvement with RLVR than the next best method (Section 4.2.1). Then, in Section 4.2.2, we demonstrate that `Bridge` also improves the output set quality across several metrics in terms of coverage and correctness consistency. Finally, in Section 4.3, we highlight some important characteristics of our method including the versatility of generation width, length extrapolation, benefit of warm up, feature contributions, and output stability.

### 4.1 EXPERIMENTAL SETTINGS

**Models and Baselines.** We test `Bridge` on DeepSeek-R1-Distill-Qwen-1.5B, DeepSeek-R1-Distill-Qwen-7B, and DeepSeek-R1-Distill-Llama-8B, which we abbreviate to DS-Qwen-1.5B, DS-Qwen-7B, and DS-Llama-8B, respectively (Dubey et al., 2024; Yang et al., 2024; Guo et al., 2025). We use 4 query and key-value attention heads for `Bridge`, each with the same dimension as the original model's head dimension. This only adds 5.1%, 2.8%, and 3.4% extra parameters on top of the original DS-Qwen-1.5B, DS-Qwen-7B, and DS-Llama-8B models, respectively. Table 7 in Appendix B lists the exact parameter counts. Our parameter-matched baseline which we call "P-Match" adds 2-layer MLPs of the same size in the same positions as `Bridge` blocks which serves to show the limited effect of just adding parameters. Matched in parameter count, P-Match and `Bridge` are also trained with the same warm up and RLVR pipeline. Both methods are initialized to have zero contribution.

**Training.** For the SFT warm up stage, we first use the original LLM to generate 8 response for each GSM8K (Cobbe et al., 2021) problem and then filter out incorrect responses and problems with one or fewer correct responses. We train only the additional parameters with `Bridge` and P-Match on this custom dataset for 5 epochs and keeping the best checkpoint according to the perplexity on 500 validation problems (and their corresponding set of correct reasoning traces). This checkpoint is inserted in the model for RLVR where we train the full model on DeepScaleR-Preview-Dataset (Luo et al., 2025) for 1000 gradient steps. DS-Qwen-1.5B is trained with generation width 8 while the others were trained with 4. The only reward is correctness of the generation. Our training hyperparameters are listed in Appendix A.

**Evaluation.** We evaluate `Bridge` on 7 math benchmarks (MATH-500 (Hendrycks et al., 2021; Lightman et al., 2023), AIME24, AIME25 (AIME, 2025), AMC23 (AMC, 2023), BRUMO25 (BRUMO, 2025), CMIMC25 (CMIMC, 2025), and HMMT_FEB25 (HMMT, 2025)) and 5 challenging non-math benchmarks (XSum (Narayan et al., 2018), CNN/DailyMail (Hermann et al., 2015; See et al., 2017), GPQA (Rein et al., 2024), ZebraLogic (Lin et al., 2025), and Countdown (Pan et al., 2025b)). Evaluating MATH-500 on every 100 training steps, the checkpoint with the highest validation accuracy is used to test on the remaining benchmarks. We evaluate across 4 responses per MATH-500 sample and 32 responses per sample from the other benchmarks. Sampling temperature and top-$p$ are set to 0.6 and 0.95, respectively. We set the generation width of `Bridge`

Table 1: Accuracy comparison across math benchmarks. In each section, the 4 rows from top to bottom are the performance of the original model, RLVR applied on the original model, P-Match (extra MLPs) with SFT warm up and RLVR, and `Bridge` with SFT warm up and RLVR. The 2 rightmost columns show the average across all benchmarks and the average improvement over the original model. MATH-500, AMC23, BRUMO25, CMIMC25, and HMMT_FEB25 are abbreviated to MATH, AMC, BRU, CMI, and HMMT, respectively.

| Model | MATH | AIME24/25 | AMC | BRU | CMI | HMMT | Avg | $\uparrow \Delta$ |
|---|---|---|---|---|---|---|---|---|
| *DS-Qwen-1.5B* | 73.65 | 13.75 / 13.44 | 50.00 | 18.12 | 4.30 | 8.23 | 25.93 | 0.00 |
| RLVR only | 78.75 | 17.40 / 18.44 | 60.55 | 18.54 | 3.83 | 7.50 | 29.29 | 3.36 |
| P-Match | 78.65 | 18.12 / 19.17 | **60.62** | 20.94 | 5.08 | 8.54 | 30.16 | 4.23 |
| `Bridge` | **81.30** | **20.11 / 20.00** | 60.55 | **21.36** | **5.63** | **9.79** | **31.25** | **5.32** |
| *DS-Qwen-7B* | 82.15 | 23.44 / 21.88 | 66.02 | 23.75 | 5.63 | 11.98 | 33.55 | 0.00 |
| RLVR only | **88.15** | 29.06 / 23.85 | 74.30 | 28.33 | 7.97 | **12.60** | 37.75 | 4.20 |
| P-Match | 86.80 | 28.85 / **25.73** | 70.47 | 26.77 | 6.25 | 11.87 | 36.68 | 3.13 |
| `Bridge` | **88.15** | **32.19** / 25.41 | **77.65** | **30.21** | **9.77** | 12.40 | **39.40** | **5.85** |
| *DS-Llama-8B* | 73.40 | 15.42 / 13.12 | 57.97 | 15.62 | 2.73 | 8.23 | 26.64 | 0.00 |
| RLVR only | 76.70 | 18.12 / 18.12 | 63.44 | 15.83 | 5.47 | 10.52 | 29.74 | 3.10 |
| P-Match | 78.00 | 22.29 / **20.21** | 61.80 | 17.81 | 5.08 | 11.67 | 30.98 | 4.34 |
| `Bridge` | **80.15** | **24.76** / 18.18 | **66.36** | **19.91** | **6.02** | **11.93** | **32.47** | **5.83** |

to 8 for all tasks except MATH-500, which we set to 4 since we only evaluate on 4 responses per sample. We adapt our evaluations from the Lighteval framework (Habib et al., 2023).

## 4.2 REASONING PERFORMANCE

Here, we show the performance improvements of our method `Bridge` which leverages inter-sample information sharing for high quality generations. We evaluate performance both on per-output accuracy (Section 4.2.1) and macroscopically, on the set of outputs generated per prompt (Section 4.2.2).

### 4.2.1 ACCURACY

Beginning with standard accuracy (Pass@1), we compare the performance of the original model, original model with RLVR, P-Match with SFT and RLVR, and `Bridge` with SFT and RLVR on several math benchmarks. Results in Table 1 show that in nearly all cases and on average, `Bridge` obtains the highest accuracy compared to all other methods. *In particular, the average performance improvements of our method on the original model is 26%, 39%, and 34% relatively more than that of the next best method on DS-Qwen-1.5B, DS-Qwen-7B, and DS-Llama-8B models, respectively.* P-Match with parameter counts pegged to `Bridge` improves accuracy from just pure RLVR most of the time but is much more inconsistent, such as in the case of DS-Qwen-7B. This indicates that the superior performance of `Bridge` is not solely attributed to additional parameters. Furthermore, even though DS-Qwen-7B and DS-Llama-8B were trained with generation width 4, the evaluation results with width 8 are still stronger than the other independent sampling methods, showing the robustness of `Bridge`. In addition, the improvement by `Bridge` is greater for larger models, and scaling up to even larger ones remains of interest for future work. Although we train `Bridge` solely on math, we observe no degradation and sometimes improvement on non-math tasks (Table 2).

### 4.2.2 SET EVALUATIONS

Zooming out, we show `Bridge` also improves the consistency and coverage (i.e., the percentage of questions that have at least 1 correct response in the response set) across multiple generation attempts. To evaluate the set of responses to a single input, we use the G-Pass@$k_\tau$ (Liu et al., 2025) metric, which paints a more holistic picture of model potential (coverage) and consistency. Whereas Pass@$k$ is the probability of a correct output in $k$ responses, G-Pass@$k_\tau$ is the probability of $0 < \tau \le 1$ fraction of $k$ responses being correct. More formally, for $n$ responses and $c$ correct

Table 2: Evaluations on non-math tasks. Note that our training procedure only used math samples. Rouge-1 (Lin, 2004) scores are reported for summarization (XSum and CNN/DailyMail). Average accuracies are reported for GPQA, ZebraLogic, and Countdown.

| Model | XSum | CNN/DailyMail | GPQA | ZebraLogic | Countdown |
|---|---|---|---|---|---|
| *DS-Qwen-1.5B* | 15.72 | 22.11 | 33.14 | 30.90 | 28.77 |
| RLVR only | 14.81 | 22.79 | 32.45 | 30.90 | 28.15 |
| P-Match | 15.90 | 22.78 | 32.51 | 32.55 | 31.36 |
| Bridge | **17.17** | **24.07** | **33.90** | **33.15** | **34.84** |
| *DS-Qwen-7B* | 18.03 | 24.19 | 43.94 | 40.00 | 49.55 |
| RLVR only | 17.24 | 23.52 | 43.56 | 41.25 | 49.93 |
| P-Match | 18.13 | 23.76 | 43.75 | **42.95** | 46.91 |
| Bridge | **18.16** | **24.55** | **45.77** | 42.60 | **52.70** |
| *DS-Llama-8B* | 18.23 | **23.84** | 35.80 | 41.25 | 14.04 |
| RLVR only | 2.25 | 1.65 | 39.46 | 43.25 | 29.23 |
| P-Match | **19.67** | 23.01 | 38.83 | 43.50 | 32.32 |
| Bridge | 18.04 | 22.64 | **39.65** | **44.70** | **32.51** |

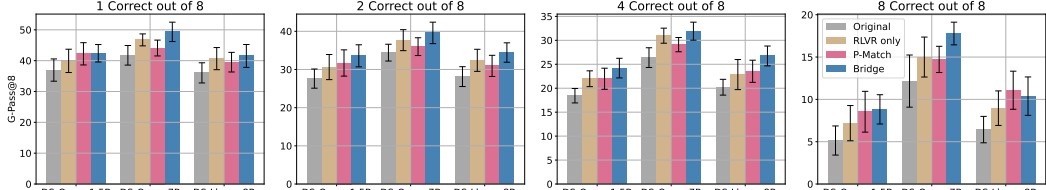

Figure 4: G-Pass@$8_\tau$ averaged across AIME24, AIME25, AMC23, BRUMO25, CMIMC25, and HMMT_FEB25. Each chart measures the minimum number of correct answers ($\tau \cdot k$) out of $k = 8$ simultaneous responses. Bridge has the greatest coverage ($\tau \cdot k = 1$) and answers correctly most consistently ($\tau \cdot k > 1$) in the vast majority of cases. Higher is better.

responses,

$$\text{Pass@}k = \mathbb{E}\left[1 - \frac{\binom{n-c}{k}}{\binom{n}{k}}\right], \qquad \text{G-Pass@}k_\tau = \mathbb{E}\left[\sum_{j=\lceil \tau k \rceil}^{c} \frac{\binom{c}{j} \cdot \binom{n-c}{k-j}}{\binom{n}{k}}\right].$$

As $\tau \to 0$, G-Pass@$k_\tau$ is simply the coverage. On the other extreme, G-Pass@$k_1$ is the probability that all $k$ responses are correct.

From Figure 4, Bridge achieves higher G-Pass@$8_\tau$ values for nearly all values of $\tau$ and models. This demonstrates that Bridge can achieve greater coverage without spreading out its responses to many incorrect answers. *In other words, not only do Bridge blocks increase the probability of a correct response in the response set more than the other methods, they also increase the frequency at which they occur.* Again, we note that Qwen-7B and DS-Llama-8B were trained with generation width 4 yet they generalize well to evaluation width 8.

### 4.3 ABLATIONS AND ANALYSIS

**Generation Width.** The design of Bridge allows complete flexibility in the number of parallel generations, or generation width $w$, due to the removal of positional encoding. Here, we show its generalizability to other widths on DS-Qwen-7B which was trained on a width of 4 with RLVR. In Table 3, in all cases where $w > 1$, Bridge outperforms P-Match in terms of task-wise and global average accuracy. We also investigate the effect of $w$ on set quality in Figure 5. Again, we generally see a vast improvement upon the original model and P-Match with $w > 1$ for all G-Pass@$8_\tau$ settings. *These results show not only the benefit of sharing information via Bridge but also the generalizability to widths wider and thinner than its training width.* At the extreme of $w = 1$, equivalent to independent generations, results in average accuracy that falls between RLVR only and P-Match, indicating that Bridge blocks do not harm independent reasoning.

Table 3: Accuracy across 32 samples of varying `Bridge` generation widths, $w$, with DS-Qwen-7B which was trained at width 4 with RLVR. A `Bridge` width of 1 is equivalent to independent generation. Tasks are abbreviated as described in Table 1.

| Method | AIME24 | AIME25 | AMC | BRU | CMI | HMMT | Avg | $\uparrow \Delta$ |
|---|---|---|---|---|---|---|---|---|
| *DS-Qwen-7B* | 23.44 | 21.88 | 66.02 | 23.75 | 5.63 | 11.98 | 25.45 | 0.00 |
| RLVR only | 29.06 | 23.85 | 74.30 | 28.33 | 7.97 | 12.60 | 29.35 | 3.90 |
| P-Match | 28.85 | **25.73** | 70.47 | 26.77 | 6.25 | 11.87 | 28.32 | 2.87 |
| `Bridge` ($w = 1$) | 28.13 | 24.48 | 74.85 | 28.02 | 9.07 | 11.77 | 29.39 | 3.94 |
| `Bridge` ($w = 4$) | 31.57 | 25.63 | 76.93 | 28.65 | **10.16** | **13.13** | 31.01 | 5.56 |
| `Bridge` ($w = 8$) | 32.19 | 25.41 | **77.65** | **30.21** | 9.77 | 12.40 | **31.28** | **5.82** |
| `Bridge` ($w = 16$) | **32.92** | 25.11 | 75.70 | 30.63 | 8.21 | 12.50 | 30.85 | 5.40 |

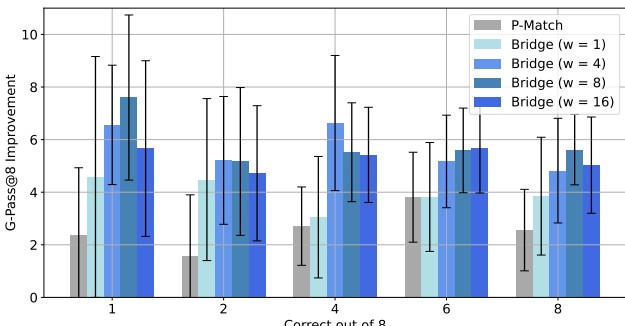

Figure 5: G-Pass@$8_\tau$ improvement upon the original DS-Qwen-7B model averaged across AIME24, AIME25, AMC23, BRUMO25, CMIMC25, and HMMT_FEB25 with relation to the evaluation generation width $w$ of `Bridge`. The x-axis ($\tau \cdot k$) indicates the number of responses out of $k = 8$ that must be correct.

**Generation Length.** `Bridge` also shows strong generalizability along the length axis. We demonstrate its performance as we extrapolate beyond its training length of 4096, again measuring both individual and set performance. From Figure 6, our method scales smoothly and better than the other baselines in most cases. At the individual response level, our method achieves the highest accuracy across all generation lengths. At the set level, `Bridge` blocks increase the number of sets that *only* had correct answers by 6.0% compared to the next best at 16K generation length, illustrating our method's consistency to generate correct answers.

**Cold Start vs. Warm up.** Warming up `Bridge` blocks with SFT prior to RLVR outlined in Section 3.2 leads to improvements in performance, shown in Table 4. The slight improvement implies that although it is prefered to warm up these new layers, it is not catastrophic if RLVR is applied directly from initialization.

Table 4: Accuracy comparison between cold start RLVR and RLVR with SFT warmed up `Bridge` blocks in DS-Qwen-7B. Tasks are abbreviated as described in Table 1. 16 responses were collected per task except MATH-500 in which 4 were collected.

| | MATH | AIME24/25 | AMC | BRU | CMI | HMMT | Avg |
|---|---|---|---|---|---|---|---|
| Cold Start | **88.70** | 33.13 / **26.67** | 77.19 | 29.80 | 7.04 | 12.09 | 39.23 |
| Warmed up | 88.15 | **33.75** / 25.63 | **77.97** | **30.21** | **10.00** | **13.13** | **39.83** |

**Feature Contribution.** Having shown the improved performance brought by `Bridge`, we now briefly peer into the effect that it has on LLM hidden states. We measure this by finding the ratio between the output norm of each block with the corresponding residual norm of each token, with lower values suggesting relatively little effect on the residual features (Figure 7). Surprisingly, we find `Bridge` blocks contribute little compared to its counterpart in P-Match, despite having a significant impact on the performance.

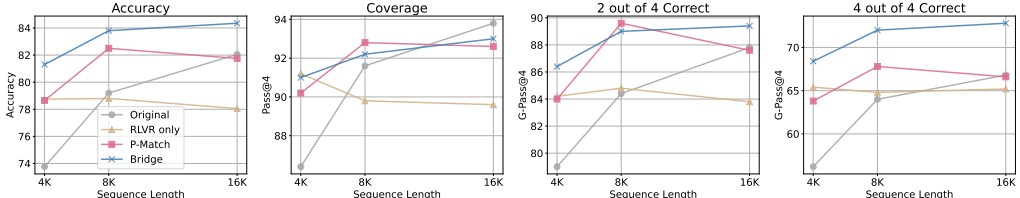

Figure 6: From left to right, DS-Qwen-1.5B MATH-500 accuracy, coverage, G-Pass@$4_{0.5}$, and G-Pass@$4_1$ as generation length increases. We generate 4 responses per input.

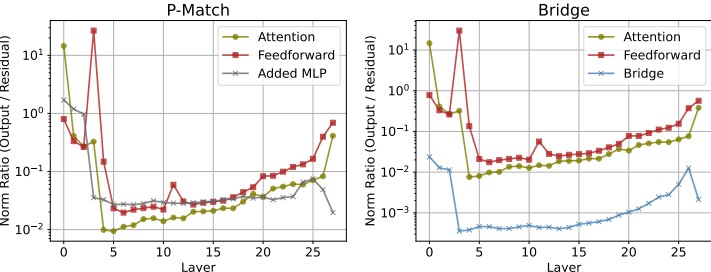

Figure 7: Ratio between feature norms of the block output and residual of every DS-Qwen-7B layer.

**Output Stability.** We additionally measure the effect of `Bridge` on the output tokens in Table 5. First, we find the average pair-wise BERTScores (Zhang et al., 2019) between MATH-500 responses, where higher scores indicate more similar output sequences. Our method has a slightly higher BERTScore, meaning `Bridge` marginally increases output similarity but crucially does not collapse the distribution of outputs. Second, we measure the variance in the evaluation results for different responses to the same prompt. For this, we turn to summarization tasks where the evaluation metric (Rouge) of a single response is more fine-grained than the binary nature of math tasks. With the lowest variance, `Bridge` produces outputs with the most consistent quality.

Table 5: DS-Qwen-7B BERTScores (F1) for MATH-500 and Rouge-1 variances of summarization tasks. Higher BERTScores indicate greater similarity of outputs.

|            | MATH-500 BERTScore | CNN/DailyMail Variance | XSum Variance |
|------------|--------------------|------------------------|---------------|
| *DS-Qwen-7B* | 89.93            | 16.14                  | 21.21         |
| RLVR only  | 90.06              | 27.96                  | 33.15         |
| P-Match    | 89.89              | 16.09                  | 22.29         |
| `Bridge`   | 90.41              | 14.44                  | 20.23         |

## 5 CONCLUSION

To generalize and enhance parallel inference scaling for LLMs, we introduce `Bridge`, a novel and inexpensive architectural addition to LLMs that allows parallel generations for the same input to share information with each other throughout the decoding process. We demonstrate that our method improves both single sample accuracy and set-wise quality across multiple models and several reasoning tasks. We achieve this by rethinking hidden states in parallel scaling as higher order tensors rather than disjoint slices. With this interpretation, this also plants the seeds for many exciting future directions such as observing the the effect of `Bridge` blocks during pretraining or mid-training, post-training with a Pass@$k$ (Chen et al., 2025b) or another global objective, and quantifying the benefit on other modalities and tasks which can exhibit different levels of output homogeneity (Jain et al., 2025). Such directions will push parallel scaling as a much more effective axis of LLM inference scaling.

ACKNOWLEDGMENTS

We thank Bradley Brown for frequent discussions about this project.

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

## A  TRAINING HYPERPARAMETERS

Table 6 lists the hyperparameters used for RLVR and SFT.

Table 6: Training hyperparameters.

| Hyperparameter | RLVR | SFT & Warmup Data Generation |
|---|---|---|
| Mixed precision | BF16 | BF16 |
| Optimizer | AdamW | AdamW |
| KL penalty | 1e-3 | N/A |
| Learning rate | 1e-6 | 2e-5 |
| LR scheduler | constant | linear |
| LR warmup | 10% | 10% |
| Training steps | 1000 | varies |
| Batch size (rollouts $\times$ samples) | 448 | 32 |
| Rollouts per sample | 8 | [2,8] |
| Total samples | 7000 | varies |
| Max length | 4096 | 2048 |
| Steps per rollout | 8 | N/A |
| Temperature | 0.6 | 0.6 |
| Heads | 4 | 4 |
| Generation width | 4 or 8 | [2,4] |

## B  PARAMETER COUNT BREAKDOWN

`Bridge` has a low memory cost, adding relatively very few parameters (2.8% to 5.1%) to LLMs. Table 7 shows the exact parameter counts for each model.

Table 7: Distribution of parameters (B) across embedding/head, attention (Attn), feedforward (FF), and `Bridge` blocks.

| Model | Embed & Head | Attn | FF | `Bridge` | Orig. Total | New Total |
|---|---|---|---|---|---|---|
| DS-Qwen-1.5B | 0.47 | 0.15 | 1.16 | 0.09 | 1.78 | 1.86 |
| DS-Qwen-7B | 1.09 | 0.82 | 5.70 | 0.21 | 7.62 | 7.82 |
| DS-Llama-8B | 1.05 | 1.34 | 5.64 | 0.27 | 8.03 | 8.30 |

## C  FURTHER GRPO CONSIDERATIONS

While our method inserts dependence between sequence and therefore their corresponding rewards, the sample permutation invariance of `Bridge` blocks means the unconditional rewards are still identically distributed. This implies

$$\mathbb{E}(\hat{A}_i) = \mathbb{E}\left(\frac{r_i - \frac{1}{n}\sum_{j=1}^{G} r_j}{\text{std}(r_1,\ldots,r_G)}\right) = \mathbb{E}\left(\frac{r_i - \frac{1}{n} \cdot n r_i}{\text{std}(r_1,\ldots,r_G)}\right) = 0,$$

preserving unbiasedness of the advantage. To preserve some notion of independence between rollouts for GRPO, one can generate multiple groups per prompt with `Bridge` and compute advantages between groups. Though this deserves exploration in future work, we do not do this here as it would be computationally expensive, and our single group setup is already empirically performative.

## D  BRIDGE PLACEMENT

Here, we examine in the architectural placement of `Bridge` blocks. In Table 8, we compare the resulting MATH500 accuracy after applying RLVR on DS-Qwen-1.5B with `Bridge` blocks added

after attention blocks or after feedforward blocks, our chosen architecture for the experiments. Since there is not a significant difference, this implies flexibility in placement, though we choose to stick with the one with the higher warmed up performance for our experiments.

Table 8: Effect on MATH-500 accuracy when inserting `Bridge` blocks after attention blocks vs. after feedforward blocks (chosen architecture) in DS-Qwen-1.5B.

| Placement | Cold Start | Warmed up |
|---|---|---|
| After Attention | 80.20 | 80.20 |
| After Feedforward | 80.15 | 81.30 |

## E    SELF-CONSISTENCY

Since `Bridge` solely focuses on generation, we can use any post-generation method to synthe-size or choose the final output. As an example, Table 9 demonstrates `Bridge` coupled with self-consistency (Wang et al., 2022).

Table 9: Self-consistency accuracy across 32 samples. Tasks are abbreviated as described in Table 1.

| Model | AIME24 | AIME25 | AMC | BRU | CMI | HMMT | Avg |
|---|---|---|---|---|---|---|---|
| *DS-Qwen-1.5B* | 20.83 | 19.17 | 66.87 | 30.83 | 13.75 | 10.83 | 27.05 |
| RLVR only | 28.33 | 25.00 | 71.25 | 24.17 | 7.50 | 13.33 | 28.26 |
| P-Match | 25.00 | 26.67 | 72.50 | 28.33 | 8.75 | 15.00 | 29.38 |
| Bridge | 30.83 | 24.17 | 75.00 | 28.33 | 11.25 | 15.00 | 30.76 |
| *DS-Qwen-7B* | 30.00 | 26.67 | 78.12 | 33.33 | 8.75 | 20.83 | 32.95 |
| RLVR only | 36.67 | 27.50 | 83.75 | 38.33 | 14.37 | 17.50 | 36.35 |
| P-Match | 37.50 | 29.17 | 81.87 | 36.67 | 13.13 | 15.00 | 35.56 |
| Bridge | 40.00 | 30.00 | 86.25 | 41.67 | 17.50 | 16.67 | 38.68 |
| *DS-Llama-8B* | 25.00 | 16.67 | 73.12 | 23.33 | 8.13 | 13.33 | 26.60 |
| RLVR only | 25.83 | 25.00 | 76.25 | 25.83 | 12.50 | 17.50 | 30.49 |
| P-Match | 25.83 | 25.00 | 70.63 | 25.83 | 14.37 | 19.17 | 30.14 |
| Bridge | 31.67 | 22.50 | 78.75 | 24.17 | 10.00 | 15.00 | 30.35 |

## F    BRIDGE PSEUDOCODE

Algorithm 1 sketches the pseudocode for `Bridge` blocks, following (2). Like normal self-attention, this can easily be extended to multiple heads.

---

**Algorithm 1** `Bridge` Block

---

**Input:** $\mathcal{X} \in \mathbb{R}^{B \times S \times D}$
**Parameters:** $\boldsymbol{W}_{\mathrm{Q}}, \boldsymbol{W}_{\mathrm{K}} \in \mathbb{R}^{D \times D_{\mathrm{QK}}}; \boldsymbol{W}_{\mathrm{V}}, \boldsymbol{W}_{\mathrm{O}}^{\top} \in \mathbb{R}^{D \times D_{\mathrm{VO}}}$
**Output:** $\mathcal{Y} \in \mathbb{R}^{B \times S \times D}$
$\boldsymbol{Q}_s \leftarrow [\mathcal{X}]_{\cdot,s,\cdot} \boldsymbol{W}_{\mathrm{Q}}$ for $s = 1, \ldots, S$
$\boldsymbol{K}_s \leftarrow [\mathcal{X}]_{\cdot,s,\cdot} \boldsymbol{W}_{\mathrm{K}}$ for $s = 1, \ldots, S$
$\boldsymbol{V}_s \leftarrow [\mathcal{X}]_{\cdot,s,\cdot} \boldsymbol{W}_{\mathrm{V}}$ for $s = 1, \ldots, S$
Construct mask $\boldsymbol{M}_s \in \mathbb{R}^{B \times B}$ for $s = 1, \ldots, S$:
  $[\boldsymbol{M}_s]_{b_1, b_2} = 0$ if generations $b_1, b_2$ have the same prompt and are incomplete at token $s$.
  $[\boldsymbol{M}_s]_{b_1, b_2} = -\infty$ otherwise.
$[\mathcal{Y}]_{\cdot,s,\cdot} \leftarrow \mathrm{Softmax}\left(\frac{\boldsymbol{Q}_s \boldsymbol{K}_s^{\top}}{\sqrt{D_{\mathrm{QK}}}} + \boldsymbol{M}_s\right) \boldsymbol{V}_s \boldsymbol{W}_{\mathrm{O}}$ for $s = 1, \ldots, S$
**Return:** $\mathcal{Y}$

---

