# OpenReview forum: "Generalized Parallel Scaling with Interdependent Generations"
_ICLR.cc/2026/Conference — ICLR 2026 Poster_

### Official Review · Reviewer_hSjT · 2025-10-30

**Soundness:** 3
**Presentation:** 4
**Contribution:** 3
**Rating:** 8
**Confidence:** 3

**Summary:**

Their method allows the LLM to attend to parallel generations from the same prompt. They do this with a new transformer block that attends to the previous token representation from the other generations. They initialize the layers to have no contribution, and fine tune the new architecture with SFT followed by RL. Their results show that this works better than adding the equivalent number of parameters to the transformer.

**Strengths:**

Novelty:
Existing methods use multiple LLM generations, but theirs is the first I know of in which other generations influence the model during generation.

Clarity:
The paper was easy to understand, Figure 2 was clear and aided in understanding.

Results:
The results are strong. I found figure 5 especially convincing, since it shows that increasing the generation width helps up to a point.

Significance:
It's common to sample multiple responses from LLMs, so this method is widely applicable.

**Weaknesses:**

Error bars

None of the plots have error bars and there’s only one run of each method. This makes it hard to tell if the differences are real in some cases where the method results are close to the baseline.

Reproducibility

The code is not provided, and there is no reproducibility statement. This is especially important for this paper since the main contribution is the new transformer layer so open sourcing the implementation would be very useful.

Baselines

They don’t compare with any existing methods. Some existing methods take advantage of multiple generations to improve LLM accuracy, for example self consistency. The authors could compare against self consistency, or show the result of using self consistency on top of Bridge.

Minor clarity issues:

The motivation in the introduction is high level and abstract. It says “Independent generations for the same prompt leave potentially useful information derived from other responses unutilized, limiting the performance ceiling.” but it doesn’t explain what this potentially useful information is. Add concrete examples to help the reader understand the motivation.

The paper doesn’t explain what RL with verifiable rewards is, leading to confusion which don’t already understand it. For example, the paper should explain if Bridge can also work with other types of RL for language models like RLHF, or if it’s only applicable to RLVR.

**Questions:**

none

---

> ### Author Response · Authors · 2025-11-20
>
> Thank you for the thorough review of our paper! Below, we provide clarifications on each of your comments in order. Please let us know if you have any additional comments!
>
> **Error Bars**
>
> Thank you for the suggestion, we have added them to our plots in our revised version. Note for all tasks except MATH-500, we generated 32 responses per problem.
>
>
> **Reproducibility**
>
> Of course, we plan to release code for our architecture, training, and evaluation as soon as we finish cleaning up the codebase.
>
>
> **Baselines**
>
> See Bridge and self-consistency results in Table R2. Please note that Bridge and self-consistency are _not_ replacements of each other since Bridge can work with any post-generation synthesis method. Even so, self-consistency results look encouraging.
>
> To comment more on comparisons, existing parallel methods use N parallel threads to generate 1 response (N-to-1), whereas Bridge and P-Match use N parallel threads to generate N responses (N-to-N). Hence, this is a huge difference in compute utilization when we want to generate a set of complete responses. Moreover, Bridge does not replace any of these N-to-1 methods since they can be composed with Bridge. Therefore, we think P-Match is the most appropriate baseline as we exactly match the parameter count and depth of Bridge.
>
> Table R2: Self-consistency evaluations
>
> | Model| AIME24| AIME25|AMC23|BRUMO25|CMIMC25|HMMT25|Avg|
> | - |:---:|:--:|:--:|:--:|:--:|:--:|:--:|
> |DS-Qwen-1.5B|20.83|19.17|66.87|30.83|13.75|10.83|27.05|
> | RLVR only|28.33|25.00|71.25|24.17|7.50|13.33|28.26|
> | P-Match|25.00|26.67|72.50|28.33|8.75|15.00|29.38|
> | Bridge|30.83|24.17|75.00|28.33|11.25|15.00|30.76|
> |
> |DS-Qwen-7B|30.00|26.67|78.12|33.33|8.75|20.83|32.95|
> | RLVR only|36.67|27.50|83.75|38.33|14.37|17.50|36.35|
> | P-Match|37.50|29.17|81.87|36.67|13.13|15.00|35.56|
> | Bridge|40.00|30.00|86.25|41.67|17.50|16.67|38.68|
> |
> |DS-Llama-8B|25.00|16.67|73.12|23.33|8.13|13.33|26.60|
> | RLVR only|25.83|25.00|76.25|25.83|12.50|17.50|30.49|
> | P-Match|25.83|25.00|70.63|25.83|14.37|19.17|30.14|
> | Bridge|31.67|22.50|78.75|24.17|10.00|15.00|30.35|
>
>
> **Clarity**
>
> - While the notion of interdependence is fairly abstract, this is because we let the LLM learn to share information in the latent space, rather than in the token space. If you are interested in the effect of Bridge connections, we show in Table R1, Bridge performs some sort of variance reduction in both the answer and tokens.
>
> - Because Bridge is just an architectural change, it should work for RLHF. We thank you for bringing up this important point and have added this at the end of Section 3.3.

---

### Official Review · Reviewer_YB53 · 2025-10-30

**Soundness:** 1
**Presentation:** 2
**Contribution:** 3
**Rating:** 2
**Confidence:** 4

**Summary:**

The paper proposes Bridge, a module for LLMs that applies attention between (past) tokens across parallel samples. The proposed method enables interdependence between traditionally independent parallel responses. The block is trained using RLVR and optional SFT warm up. The paper presented experiments showing that Bridge achieves better accuracy on math benchmarks over the base model.

**Strengths:**

- The paper presents, to the best of my knowledge, a novel idea of enabling attention between tokens across samples in a batch, a dimension that typically maintains independence.
- The evaluation applies Bridge and baselines to a variety of different math benchmarks

**Weaknesses:**

- The improvement in accuracy is rather small compared to baselines. Bridge does not perform much better than P-Match, which is the baseline given the equivalent amount of compute and in some cases also not much better than just RLVR only. As such, it seems most of the accuracy improvements are from additional compute/training over the original model rather than from the Bridge block design.
- An important motivation to enable parallelism is to enable lower latency while increasing compute. The evaluation does not present the inference latency of Bridge and baselines
- The evaluation lacks comparison to methods that enable parallelism during LLM decoding (such as those cited in the paper). Several of those methods (e.g. Multiverse, Pasta) already have mechanisms for encoding interdependency between different threads by enabling parallel threads to join together and respawn. The evaluation should evaluate against these methods to compare how Bridge's method of enabling interdependency performs.
- The notation of Eq 1 and Eq 2 are difficult to parse

**Questions:**

- Is there a latency cost due to the additional computation in Bridge?
- How does Bridge perform compared against using existing methods enabling parallel computation (with and without interdependence)? Even though they eventually join together into a single response, a simple string post-processing step can split the response back into a set of responses by splitting on the special tokens these methods use for spawning parallel threads. How does Bridge compare to doing this?

---

> ### Author Response · Authors · 2025-11-20
>
> Thank you for the constructive comments and questions! We address each point you brought up in the bullets below. Since there is overlap with some of your comments and questions, we combined them to address them together. Please let us know if we have addressed them appropriately.
>
> **Performance/Contribution**
>
> _To help build confidence in our performance, we have doubled the number of responses per test sample from 16 to 32. These new numbers are reflected in our revision and further support our claim._ Most of the math tasks are difficult competition-level questions, meaning that the differences in accuracy on these tasks are significant. We see that RLVR only and P-Match are improving average accuracy by 3-4.5%, showing that additional parameters may not necessarily improve performance. On the other hand, Bridge can improve average accuracy by close to 6% which, relative to the gains made by other methods, is significant. Moreover, Bridge nearly always achieves the best performance at individual tasks. We would also like to emphasize that beyond performance, another core contribution is the idea of mixing information in latent space (rather than in the token space) between generations for parallel scaling to generate **sets** of responses. Hence, Bridge is a simple generalization of parallel inference by performing computation along an axis which is usually kept independent.
>
> **Latency**
>
> There is a small latency cost of our method (~8% increase for Deepseek-Distill-Qwen-7B with Bridge). While the focus of this paper is not latency, in terms of efficiency, we highlight that we have virtually no increase in memory consumption due to the small number of additional parameters and no feature storage.
>
> **Comparisons**
>
> The other parallel methods described use N parallel threads to generate 1 response (N-to-1), whereas Bridge and P-Match use N parallel threads to generate N responses (N-to-N). Hence, this is a huge difference in compute utilization when we want to generate a set of complete responses. Moreover, Bridge does not replace any of these N-to-1 methods since they can be composed with Bridge. Using special tokens to spawn N parallel threads to generate N complete responses is equivalent to independent generation. Therefore, we think P-Match is the most appropriate baseline as we exactly match the parameter count and depth of Bridge.
>
> **Notation**
>
> Thank you for bringing this up. We have cleaned up the notation of Eq 1 and 2 in our revision. Please check if this is clearer.

---

> > ### Comment · Reviewer_YB53 · 2025-11-25
> >
> > Thank you for the responses.
> >
> > Performance: My comment that Bridge does not improve much in performance compared to P-Match (the parameter matched baseline) remains. In Math, P-Match improves accuracy to the base model by 4-5%, and Bridge only improves over P-Match by ~2%. This indicates that most of the 6% improvement Bridge has over the base model is due to the additional parameters. This is also the case in AIME: P-Match improves accuracy to the base model by 4-7%, and Bridge only improves over P-Match by only 1-4%.
> >
> > Contribution: I acknowledge that the idea of mixing information in latent space is novel. However, I am not convinced the accuracy improvement for doing so is significant given that most of the improvement is from additional parameters. If the claim is that it enables producing more correct answers within a set of generated answers (Fig 4), then I am not sure that benchmarks with only a single answer is the right benchmarks. It would be more relevant for applications where you want a set of correct but different answers. Could the mixing not cause a mode collapse? Even then, in Fig 4, Bridge does not out-perform P-match for getting 8 correct out of 8 on DS-Qwen-1.5B and DS-Llama-8B.
> >
> > Latency: Could these results be included?
> >
> > Comparisons: Why does it matter if the method is N-to-1 or N-to-N if the benchmarks only require 1 answer and the primary metric is accuracy not latency? Additionally, the related works enable interdependence between the N parallel threads by allowing them to join and gain visibility to each other before re-spawning M parallel threads. As such, I believe they are still important baselines.
> >
> > Notation: Thank you, this is better.

---

> > > ### Author Response · Authors · 2025-11-25
> > >
> > > Thanks for the response! Some follow-ups on your comments:
> > >
> > > **Performance and Contribution**
> > >
> > > The purpose of P-Match is to show that additional parameters do not help much since although it adds the same number of parameters as Bridge, P-Match performance is about equivalent (and sometimes even less than in the case of DS-Qwen-7B) to RLVR only. RLVR is the most helpful to boost performance, and with Bridge, we can extend the headroom that RLVR improves performance. In other words, the difference between RLVR only and P-Match is the actual benefit of additional parameters, which is not much if at all.
> > >
> > > You are correct that average accuracy should not be the only metric to assess Bridge. We show that with evaluations at the set level, Bridge achieves the highest GPass-k in the vast majority of cases (Figures 4 and 5), though not unanimously in every case as you pointed out. To further reinforce this, we would like to recommend you to take a look at Table R2 (under Reviewer hSjT), which shows promising results when Bridge is paired with self-consistency.
> > >
> > > Since you seem curious about what Bridge connections are doing, we would also like to point you to Table R1 to show that Bridge stabilizes output generations and answers, even for tasks without concrete answers.
> > >
> > > **Latency**
> > >
> > > Of course, we can add a discussion on this.
> > >
> > > **Comparisons**
> > >
> > > Comparing N-to-1 with N-to-N is unfair due to the substantial amount of resources the former uses, especially since we sample 32 responses for most of our evaluated tasks. Not only do N-to-1 methods perform N times more computations, they also need to store KV caches N times in size. Moreover, they are not replacements of each other. Since one can add Bridge blocks into N-to-1 models which usually only modify attention blocks. Thus, we do not think they are fair comparisons due to differences in compute, storage, purpose, and design.

---

> > > > ### Comment · Reviewer_YB53 · 2025-11-26
> > > >
> > > > Thank you for the response.
> > > >
> > > > Comparisons: I am satisfied with the point that N-to-1 is fundamentally a different design paradigm to Bridge and thus incomparable. However the following concerns remain:
> > > >
> > > > Performance/Contribution: Then, the conclusion from the baselines is that most of the accuracy gain is in using RLVR and then in the additional parameters of P-Match. My concern that Bridge (the contribution of mixing information in latent space) does not significantly increase accuracy remains.
> > > >
> > > > In Fig 4 and 5, the error bars of Bridge overlap with the bars of the baselines so the result is not statistically significant.
> > > >
> > > > Table R1: The observation Bridge produces more similar outputs is not necessarily a good thing. This means that Bridge decreases the diversity and completeness of the model. Having a set of identical or similar responses does not seem useful: If the task only asks for one answer, all the responses could be all the wrong answer (in which case voting or pass@k with N responses is no longer useful). If the tasks asks for a set of different answers, the additional computation is wasted because there is only a set of the same answers.
> > > >
> > > > Tasks such as Countdown[1] might be more appropriate benchmarks if the goal is to generate a set of responses, such as to obtaining full coverage of the set of solutions to a problem.
> > > >
> > > > [1] https://huggingface.co/datasets/Parallel-Reasoning/countdown_problems

---

> > > > > ### Author Response · Authors · 2025-11-29
> > > > >
> > > > > Thank you for the quick reply!
> > > > >
> > > > > You are correct that similarity between responses is not necessarily a good thing. However, thankfully, given that we see equivalent or improved coverage (left-most graph in Figure 4 and left-most bar group in Figure 5), this means Bridge is not collapsing onto the wrong answer. These results together imply greater consistency towards the correct answer. The thinking traces are still diverse, even if they arrive at similar final answers.
> > > > >
> > > > > Thank you for suggesting Countdown! We have accuracy results on Countdown in Table R3 below, which we will include in the next paper revision. This shows strong performance for Bridge on Countdown, especially for DS-Qwen-1.5B and DS-Qwen-7B which have the highest accuracy by a large margin compared to the next best baseline. Together with Table 2, this implies Bridge has significant potential for non-math tasks, even though it was not trained beyond math.
> > > > >
> > > > >
> > > > > Table R3: Countdown accuracies.
> > > > >
> > > > > | Model|Countdown|
> > > > > | - |:---:|
> > > > > |DS-Qwen-1.5B|28.77
> > > > > | RLVR only|28.15
> > > > > | P-Match|31.36
> > > > > | Bridge|**34.84**
> > > > > |
> > > > > |DS-Qwen-7B|49.55
> > > > > | RLVR only|49.93
> > > > > | P-Match|46.91
> > > > > | Bridge|**52.70**
> > > > > |
> > > > > |DS-Llama-8B|14.04
> > > > > | RLVR only|29.23
> > > > > | P-Match|32.32
> > > > > | Bridge|**32.51**

---

### Official Review · Reviewer_T7CP · 2025-10-31

**Soundness:** 3
**Presentation:** 3
**Contribution:** 3
**Rating:** 6
**Confidence:** 2

**Summary:**

The paper proposes an architectural modification to the transformer architecture to allow parallel generations to communicate with each other during generation. It shows notable improvements in accuracy on math benchmarks.

**Strengths:**

The paper introduces a novel and clever technique that offers a nice performance improvement on math benchmarks at a modest cost. It offers an appropriate level of technical detail and analysis.

**Weaknesses:**

The paper only evaluates math benchmarks, so it's unclear how well the approach would work in other domains.

It's not clear that P-Match, rather than any of the alternatives described in the related works section, is the appropriate baseline.

Figures lack CIs.

**Questions:**

Can you address the concerns around generalization to non-math domains, and appropriate comparison approaches?

What is your hypothesis for why, as shown in Figure 7, norm ratios for Bridge blocks are so low?

---

> ### Author Response · Authors · 2025-11-20
>
> Thank you for reviewing our paper! Below, we address your comments and questions in order. Please let us know if you have additional questions!
>
> **Comments**
>
> - Other domains: **We have evaluated on 4 new non-math tasks using the same trained weights.** These include summarization (XSum and CNN/DailyMail), science (GPQA), and logic puzzle (ZebraLogic) tasks which are shown in the revised paper’s Table 2. Even though we trained purely on math, we see no degradation and often improvement on these tasks. Experimenting with a more task-heterogeneous training data mixture may be worth exploring in the future.
>
> - Baseline Choice: The other parallel methods described use N parallel threads to generate 1 response, whereas Bridge and P-Match use N parallel threads to generate N responses. Hence, this is a huge difference in compute utilization when we want to generate a set of complete responses. Moreover, Bridge does not replace any of these N-to-1 methods since they can be composed with Bridge. Therefore, we think P-Match is the most appropriate baseline as we exactly match the parameter count and depth of Bridge.
>
> - Confidence Intervals: Thank you for the suggestion, we have added them in our revision.
>
> **Questions**
>
> - Domains and baseline: See above points which address these concerns.
> - Norm Ratios: Since LLMs are not trained to perform inference by sharing features across generations, getting them to share large amounts of information may conflict the learned structure from pre-training and post-training, even though we are able to elicit some shared information here. An analogy is inducing mixture-of-experts structure after training can be more difficult than encouraging it during training. Similarly, it may be possible that by introducing Bridge earlier in the training pipeline (i.e., pretraining or mid-training) unlocks greater benefits than seen here which is an exciting avenue to explore!

---

### Official Review · Reviewer_vGTK · 2025-11-02

**Soundness:** 4
**Presentation:** 4
**Contribution:** 3
**Rating:** 6
**Confidence:** 4

**Summary:**

In this paper, the authors propose a new technique, aptly called Bridge, for post-training language models to perform well at parallel generation. By allowing different sequence positions across a batch to effectively communicate with each other, each independent generation suddenly becomes interdependent.

The authors evaluate the technique on the canonical math datasets and produce reasonable baselines to compare Bridge against. They find that Bridge leads to improvements in both the pass@k and G-Pass@8_{tau}.

**Strengths:**

Strengths:
- The paper is well written. Coherent and understandable (albeit a few minor places I would have appreciated a little more context).
- This approach seems reasonable for introducing interdependence across generations.
- Reasoning in LLMs is an important area. AFAIK this is the first paper that takes a serious stab at introducing cross information sharing across parallel generations.

**Weaknesses:**

Weaknesses:
I left the paper feeling like something was missing. The proposed architecture seems reasonable, but I have no idea for what it’s actually doing for the model?
- Does the model begin to sample in semantically different directions given context from other models? I.e., does it increase diversity?
- If one trace is suddenly going in the right direction, do other traces update their reasoning accordingly?

Something as simple as BERTScore (similar to here [1,2]) or some notion of entropy across traces in the different settings could provide some insights into what’s going on.

A preliminary step in understanding what’s going on was done on line [457] but this seems both unsatisfactory and overly short.

[1] https://arxiv.org/abs/2502.01697
[2] https://openreview.net/forum?id=gvsdQ72Peg&noteId=gvsdQ72Peg

**Questions:**

Minor points:
- How did you get the numbers 30%, 50% and 23% on line [323]? I can’t recreate them from the table using Avg col?
- In line [365] I would have appreciated a definition of coverage.

---

> ### Author Response · Authors · 2025-11-20
>
> Thank you for your comments and questions! We respond to each of your points below. Please let us know if you have any follow up questions or thoughts!
>
> **Deeper understanding of what Bridge learns**
>
> Thank you for suggesting to try measuring similarity with BERTScore. You can find these scores in Table R1. In the same table, we also measured variance in Rouge scores between responses generated together by Bridge (width of 8) and those that were generated independently. We use Rouge here because it gives us more continuity compared to the 0-1 accuracy of math problems. The performance of these added language tasks (XSum and CNN/DailyMail) are in Table 2 of the revised paper. Based on Table R1, _we observe that Bridge produces more similar outputs measured by BERTScore and reduces the variance of Rouge-1 scores, compared to all other methods._ This suggests that Bridge connections stabilize output tokens. We can add these new results into our next paper version.
>
> Table R1: Similarity measurements with response sets. For MATH-500, we find the average BERTScore (F1) between all responses in a response set. For summarization tasks, we measure the variance between the Rouge-1 scores of all responses in a response set.
>
> | Model| MATH-500 BERTScore (F1) | CNN/DailyMail Rouge-1 Variance | XSum Rouge-1 Variance |
> | --- |:---:|:----:|:---:|
> | DS-Qwen-1.5B|89.63|24.90|22.46
> | RLVR only|89.80|36.26|33.00
> | P-Match|89.85|27.12|23.76
> | Bridge|**90.43**|**20.44**|**20.68**
> |
> | DS-Qwen-7B| 89.93 | 16.14 |21.21
> | RLVR only| 90.06 | 27.96|33.15
> | P-Match| 89.89 | 16.09| 22.29
> | Bridge| **90.41** | **14.44**| **20.23**
>
>
> **Questions**
>
> 1. Thank you for trying to double-check! These percentages were taken by looking at the average performance _improvements_ (the right-most column of Table 1) and calculating (Bridge improvement)/(another method’s improvement) - 1. Namely, we show that Bridge significantly extends the _gain_ with RLVR. Note that since we have doubled the number of evaluated responses for most math tasks (see global comment for more details), the exact numbers have changed, which are also reflected in the revision.
>
> 2. Coverage is the percentage of questions that have at least 1 correct response in the response set. You can think of it as the other extreme compared to G-Pass@k_1, which is the percentage of questions with all responses correct in the response set. We have added this clarification in the revision.

---

> > ### Comment · Reviewer_vGTK · 2025-11-20
> >
> > Thanks, updated score.

---

> > > ### Author Response · Authors · 2025-11-21
> > >
> > > Thank you!

---

### Author Response · Authors · 2025-11-20
**Revision Changes**

We would like to thank the reviewers again for their time. We have made several updates to the paper in our current revision, of which we list major ones here for the interest and convenience of the reviewers:

1. 4 new non-math evaluations added to Table 2: XSum (summarization), CNN/DailyMail (summarization), GPQA (science), and ZebraLogic (puzzle). _Even though we trained purely on math, we see no degradation and often improvement on these tasks._

2. We doubled the number of responses per problem for competition math tasks from 16 to 32. As such, numbers in the tables and figures may be slightly different from the initial version. These new experiments further reinforce the benefit of Bridge.

---

### Meta-Review · Area_Chair_aemi · 2026-01-06

**Summary:**

The paper introduces Bridge, an architectural modification for Transformers that enables information sharing across parallel generations within a batch. Unlike standard parallel sampling, where sequences are independent, Bridge treats batch hidden states as holistic tensors, allowing cross-generation communication.

The primary strengths identified by reviewers are the novelty of the approach (addressing an untapped dimension of scaling), the clarity of the technical exposition, and the generality of the module, which is compatible with various post-generation aggregation techniques like self-consistency. While the initial submission focused heavily on math, the rebuttal demonstrated the method’s effectiveness across diverse domains (summarization, logic, and science).

The primary debate centered on the magnitude of improvement relative to the parameter-matched baseline (P-Match) and whether the communication induces mode collapse. While Reviewer YB53 remained skeptical regarding the performance margin, the majority of the reviewers found the results convincing, particularly the evidence that Bridge extends the headroom of RLVR gains more effectively than simply adding passive parameters.

**Reviewer Concerns:**

Addressed concerns:
- The authors provided new results on summarization (XSum, CNN/DailyMail), science (GPQA), logic (ZebraLogic), and a specific parallel reasoning task (Countdown). Bridge showed consistent improvements or stability, proving it is not overfit to math reasoning
- Reviewers questioned the comparison to N-to-1 parallel methods. The authors argued that Bridge belongs to the N-to-N paradigm (N threads for N responses), making N-to-1 methods (which use N threads for 1 response) fundamentally different in terms of compute and memory efficiency. Reviewers largely accepted P-Match as the appropriate fair baseline
- Authors added confidence intervals to figures and doubled the sample size (from 16 to 32 responses), clarifying that the gains are stable
- For reviewer YB53's question on latency, authors clarified an ~8% latency overhead for a 7B model, which reviewers YB53 found acceptable given the performance gains

Outstanding concerns:
- Reviewer YB53 maintained that the 1–4% improvement over a parameter-matched baseline (P-Match) is small. The authors argue this difference is significant in competition-level math, but this remains a point of subjective disagreement on significance
- While Bridge increases consistency (as shown by BERTScore), reviewer YB53 and vGTK remains concerned that this might reduce the diversity of the search space during sampling. However, authors provided Coverage metrics to show that the model is not collapsing on wrong answers.

**Reviewer Scores:**

Reviewer YB53 acknowledged novelty and baseline validity but likely remains a Marginal Reject due to persistent skepticism over the delta vs. P-Match, so the score might to increase from 2 to 4. For other reviewers, Reviewer vGTK explicitly raised score, and the other two reviewers' concerns were resolved and tend to maintain their original positive score unchanged.

---

### Decision · Program_Chairs · 2026-01-26

Accept (Poster)